**Subject Area:**
cellular biology/developmental biology/ genetics

*Drosophila*, genomic instability, DNA damage, cancer, aneuploidy

**Author for correspondence:**
Héctor Herranz
e-mail: hherranz@sund.ku.dk

# Genomic instability and cancer: lessons from *Drosophila*

Stephan U. Gerlach and Héctor Herranz

Department of Cellular and Molecular Medicine, University of Copenhagen, Copenhagen, Denmark

 SUG, 0000-0003-2680-0180; HH, 0000-0001-5808-1593

Cancer is a genetic disease that involves the gradual accumulation of mutations. Human tumours are genetically unstable. However, the current knowledge about the origins and implications of genomic instability in this disease is limited. Understanding the biology of cancer requires the use of animal models. Here, we review relevant studies addressing the implications of genomic instability in cancer by using the fruit fly, *Drosophila melanogaster*, as a model system. We discuss how this invertebrate has helped us to expand the current knowledge about the mechanisms involved in genomic instability and how this hallmark of cancer influences disease progression.

## 1. Introduction

Genome integrity is constantly challenged by endogenous and exogenous factors that can induce a variety of DNA lesions leading to mutations and damage in the chromosomes [1]. Each species needs to maintain the integrity of the genome to guarantee a faithful transmission of its genetic content to the next generation. To preserve the stability of genomes, different safeguard mechanisms have been developed throughout evolution. Defects in these mechanisms can lead to the accumulation of mutations, which is known as genome instability (GIN). The presence of GIN will ultimately affect gene expression and the production and activity of numerous proteins, resulting in phenotypic changes that can affect cellular fitness and trigger the development of diseases, such as cancer [2].

Cell division, one of the most fundamental traits in biology, supports development and growth, and is central to maintain homeostasis during the adult life of multicellular organisms. Cells have to copy the DNA accurately to ensure that both daughter cells acquire a complete set of the genome during mitosis. Therefore, DNA replication during S-phase is a crucial point for the maintenance of genome integrity. Aberrant replication fork progression leads to replication stress and DNA damage [3]. Oncogene activation is a source of replication stress [4–6]. This led to the oncogene-induced DNA damage model that suffices to explain how an initiating mutation can trigger a cascade of tumour-enabling factors. In this model, oncogenic activation can play a dual role: on the one hand, it causes the expansion of tumorigenic cells through higher proliferation signals, and on the other hand, it enables GIN through replication stress [7]. In addition to replicative stress, other DNA-altering factors, such as ionizing radiation, can induce DNA damage and contribute to the accumulation of mutations [8]. To combat the negative effects of DNA damage, living organisms have evolved mechanisms, generally known as the DNA damage response (DDR), to detect and repair these lesions. A defective DDR results in an accumulation of mutations that can promote the initiation and progression of diseases [9].

In a next step, dividing cells need to segregate their genetic content appropriately during mitosis to generate two daughter cells with a complete set of the genome. Accurate chromosome segregation is hence crucial to ensure cell

viability and function. However, defects in this process occasionally occur, and errors in mitosis are a source of structural and numerical chromosomal alterations commonly observed in cancer cells [10]. Mitosis is a very dynamic process. In the early stages, it involves the specification of two distinct cell poles established by centrosomes; DNA condensation into chromosomes and the formation of a mitotic spindle. This is followed by the attachment of the chromosomes to the spindle, segregation of the chromatids and, ultimately, the physical separation into two daughter cells that takes place in cytokinesis. Due to its complexity, mitosis has to be tightly coordinated. This is in large part controlled by the spindle assembly checkpoint (SAC). This mechanism monitors chromosome attachment to spindle microtubules and avoids mis-segregation by preventing cells from starting anaphase before the chromatids are connected to the spindle [11]. A malfunction of this process can result in defects in chromosome segregation known as chromosomal instability (CIN). CIN is the most frequent type of GIN in human cancers and refers to the changes in chromosome number and/or structure as a consequence of defects in mitosis. A primary consequence of CIN is the formation of cells with unbalanced chromosome content, a condition known as aneuploidy [12].

Abundant research into the mechanisms controlling the stability of genomes has revealed central insights about how defects in these processes cause genomic instability and how it contributes to tumour initiation and progression. However, we still lack a deeper understanding of the long-term effects that this hallmark of cancer has *in vivo*. The use of animal models that recapitulate the physiological consequences and the impact that these errors have in disease is crucial to elucidate the different links between GIN and cancer. This review enumerates and discusses recent findings addressing the connection between GIN and tumorigenesis using the fruit fly, *Drosophila melanogaster*, as an *in vivo* model.

*Drosophila* provides a tractable system with a sophisticated genetic toolbox that has been used to model different aspects related to human cancer [13]. It allows functional analysis *in vivo* facilitated by the accessibility to extensive collections of mutants and transgenic lines that can be used to manipulate gene activity in different contexts. Additionally, the fly genome sequence, as well as transcriptome data of different tissues and life stages, is publicly available, which supports genetic work. Flies have a short life cycle and can grow in big numbers, and fly tumours develop quickly and progress from primary tumours to malignancies in a short period of time. These factors combine to allow one to produce tumour samples on a large scale that can be used for high-throughput approaches and facilitate the generation and quick validation of hypotheses. Moreover, these tumours develop *in vivo* in an immune-proficient situation, which permits interaction with the microenvironment and with other cell types.

The first tumours in flies were identified over 100 years ago [14]. Since then, the fruit fly has been a model system used widely to study aspects of growth regulation and tumour formation [13]. For example, the cooperative interaction between the oncogene *Ras-V12* and mutants of the polarity protein *scribble* is a well-established fly cancer model that recapitulates many characteristics of human cancers, including aggressive neoplastic and metastatic behaviour [15,16]. *Drosophila* is broadly used to model different features of GIN. Here, we present the most relevant findings obtained using the fruit fly to study the connection between GIN and cancer and discuss the major advantages and limitations of modelling GIN in *Drosophila* together with the translational impact of these findings in human cancer.

# 2. Mitotic errors and chromosome instability

Flaws in mitosis are a major source of CIN that is observed in approximately 90% of solid human tumours [17,18]. Errors affecting mitosis include malfunctioning of the SAC, inefficient cohesion between sister chromatids, defective attachment between the microtubules and chromosomes, centrosome amplification, and incorrect timing of centrosome separation [19]. These defects typically result in aneuploidy. Flaws in cytokinesis, the last step of mitosis, can also occur and cause the formation of polyploid cells [20]. Below, we illustrate the strategies followed to model specific errors in mitosis and discuss the main conclusions obtained from these studies.

## 2.1. Chromosome instability in the imaginal discs

Faithful chromosome segregation is the ultimate goal during mitosis. To accomplish this, the microtubules of the mitotic spindle bind to the kinetochores—a specialized structure in the centromeric region of the mitotic chromosomes. Before the chromosomes separate in anaphase, the SAC monitors that the spindle is properly bound to the two kinetochores. Once the kinetochores are attached to the spindle, the SAC becomes inactive, which results in the activation of the anaphase-promoting complex/cyclosome (APC/C). The activation of the APC/C causes the loss of sister chromatid cohesion and transition to anaphase that will be followed by chromosome segregation and, eventually, cytokinesis [11]. In *Drosophila*, SAC dysfunction or disruption of spindle kinetochore binding causes segregation errors and aneuploidy [21–26].

The wing imaginal discs of *Drosophila* are epithelial sac-like structures present in the larva that, after metamorphosis, will give rise to the thorax and wings of the adult. This organ, formed by symmetrically dividing cells that proliferate actively during larval development, has been used extensively to model different aspects of tumorigenesis [27]. RNAi-mediated depletion of genes involved in the SAC induces CIN and aneuploidy in this organ [23,24]. Aneuploidy, a cell condition defined by an unbalanced number of chromosomes, generally has a negative effect on cell fitness [28–31]. Consistently, aneuploidy in the wing primordium leads to cell delamination and apoptosis. In mammals, the induction of apoptosis in response to CIN is p53-dependent [32,33]. In flies, this response is p53-independent and relies instead on the activation of the c-Jun N-terminal kinase (JNK) pathway [23]. Aneuploidy in the wing disc, as in mammalian cells [34], causes the production of reactive oxygen species (ROS), which contributes to the activation of the JNK cascade [22]. Suppression of apoptosis in these cells is sufficient to induce the formation of metastatic tumours [23,24] (figure 1).

## 2.2. Chromosome instability in stem cells

The response to CIN in stem cells differs notably from the one reported in symmetrically dividing cells of the imaginal

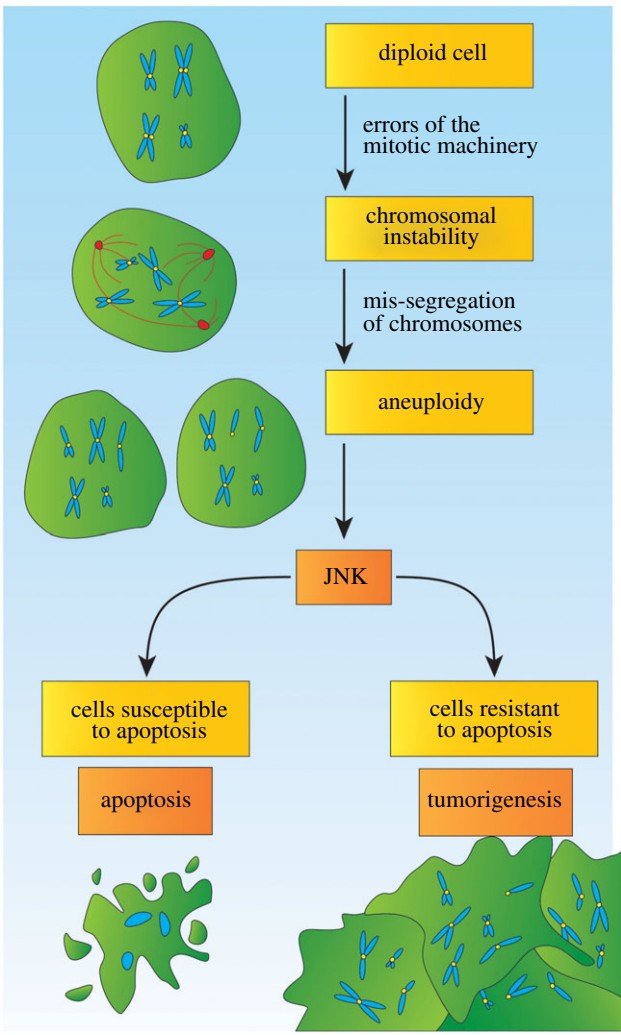

**Figure 1.** Responses to chromosomal instability. Errors of the mitotic machinery and its control mechanisms, such as the SAC, can trigger chromosomal instability. Diploid cells, including epithelial cells and stem cells, can undergo a mis-segregation of chromosomes, for example through multipolar mitosis. This can lead to aneuploidy, which describes the divergence from a diploid karyotype. In turn, the JNK stress signalling pathway is activated due to aneuploidy. This stress signalling mediates the removal of the damaged cells through apoptosis. For example, this is the case for wing imaginal epithelial cells. However, a resistance to apoptosis of cells can lead to tumorigenesis. This is the case for intestinal stem cells that generally do not undergo apoptosis. Epithelial cells that suppress apoptosis, for example through genetic manipulation, trigger tumorigenesis as well.

primordia. The *Drosophila* midgut is an excellent system to study adult stem cells *in vivo*. The maintenance and regeneration of the fly midgut and the mammalian intestine are remarkably similar and rely on the activity of intestinal stem cells (ISCs) [35–37]. Induction of CIN by SAC downregulation, kinetochore malfunction or centrosome amplification leads to the accumulation of aneuploid ISCs in the fly intestine. As observed in the imaginal epithelium, aneuploid ISCs activate the JNK pathway. However, in contrast to imaginal cells, ISCs do not undergo apoptosis and, instead, overproliferate causing the formation of tissue dysplasia [23,25]. ISCs are resistant to apoptotic signalling, and this may be central explaining the pro-tumorignic role of JNK in this specific cellular context [25,38]. In agreement with this, cultured mammalian embryonic stem cells tolerate aneuploidy and polyploidy without undergoing apoptosis [39]. Failure to remove aneuploid cells in this context might increase the levels of aneuploidy that could eventually drive tumorigenesis (figure 1).

Tissue dysplasia as a consequence of CIN does not seem to reflect a general behaviour in *Drosophila* stem cells. In the larval brain, neural stem cells, or neuroblasts, divide asymmetrically, and are essential for brain development and homeostasis [40]. Neuroblasts have the potential to stem tumour formation. For example, mutants disrupting asymmetric cell division can develop brain tumours, and these neoplasms, when injected in healthy hosts, continue growing after multiple rounds of transplantation, show malignant traits and kill eventually the host [41]. This allograft assay has been extensively used to study the oncogenic potential of different genetic conditions in flies. The analysis of mutants for the SAC components *bub3* and *bubR1* in this assay did not identify the formation of brain tumours, which indicates that the cellular responses to the presence of aneuploidies in stem cells varies depending on the cellular context [42].

## 2.3. Cytokinesis failure and polyploidy

Cytokinesis, the last step of mitosis, mediates the physical separation of the mother cell into two daughter cells. This mechanism is complex, and defects in cytokinesis can lead to the formation of polyploid cells [43]. These cells contain extra centrosomes that can disrupt normal chromosome segregation leading to the formation of aneuploid cells [44]. To limit the threat posed by this, organisms have developed tumour suppressor mechanisms. In mammals, cytokinesis failure leads to the activation of the Hippo tumour suppressor pathway, which represses the major transcriptional mediator YAP and induces the stabilization of p53. Together, these mechanisms induce G1 arrest and apoptosis [45].

We have recently developed a system to model cytokinesis failure and tumorigenesis in flies [46]. Septins play central roles in cytokinesis by controlling the ingression of the contractile ring [47]. Downregulation of the fly septin *peanut* (*pnut*) causes cytokinesis defects and tetraploidy. As observed in other fly epithelial tissues with CIN, these cells undergo apoptosis in a p53-independent and JNK-dependent manner. However, suppression of apoptosis in cells with cytokinesis failure is not sufficient to induce tumorigenesis, as has been reported in epithelia with CIN induced by other means [22–24]. Instead, cells with defective cytokinesis proliferate poorly and accumulate in G2. Overexpression of the proto-oncogene *yorkie*, the *Drosophila* homologue of mammalian YAP, is able to promote proliferation in these cells causing the formation of tumours with high ploidy cells and invasive behaviour. Functional experiments in this condition show that Yorkie exerts its pro-tumorigenic role through its target genes, the inhibitor of apoptosis *DIAP1* and the G2/M regulator *cdc25/string* [46]. To reinforce this response, Yorkie establishes a positive feedback through the induction of the microRNA *bantam*, which consolidates Yorkie activity by repressing *head involution defective* (*hid*), a proapoptotic gene that induces cell death by targeting DIAP1 and the cell cycle regulator *tribbles* (*trbl*), a pseudokinase that dampens G2/M progression by inhibiting Cdc25/string [48–55]. Hence, Yorkie suppresses apoptosis and forces G2/M progression directly, by inducing the expression of *DIAP1* and *cdc25/string* respectively, and indirectly,

through the microRNA *bantam*. MicroRNAs are proposed to be embedded in gene networks that, in normal conditions, could serve to manage biological noise and provide robustness. However, in oncogenic situations, this may boost tumorigenic programmes and promote disease progression [56–58]. The conserved microRNA *miR-8* also promotes tumorigenesis in cooperation with the oncogene EGFR [59]. Cytokinesis failure is an early event in these tumours, which occurs as a consequence of *pnut* repression by *miR-8*. This, together with EGFR upregulation, leads to the formation of giant tumour cells that use cell competition to remove surrounding cells through cell engulfment and apoptosis [59,60]. A similar cooperation has been reported between *miR-8* and Yorkie [61].

Endoreplication can be used as an alternative to mitosis. It involves a particular type of cell cycle where alternate rounds of G–S result in the formation of large cells with a single polyploid nucleus [62]. Endoreplication has been associated with oncogenic transformation and tumour progression. However, it is not well determined whether there is a causative association between both processes [63]. A recent study in flies sheds light on the implications of endoreplication in epithelial tumours. Endocytic genes, including *rab5*, are well-recognized tumour suppressors in *Drosophila* [64]. Cells depleting *rab5* activate JNK and Yorkie, which mediate a switch to endoreplication causing the formation of polyploid cells. JNK and Yorkie trigger this response by regulating the death inhibitor DIAP1 and the G2/M regulator Cyclin B [65]. Studies in flies have revealed that the inhibition of apoptosis and the induction of G2/M transition are crucial steps towards tumorigenesis in the context of endoreplication and cytokinesis failure [46,65]. Ultimately, the connection between JNK activation, commonly observed in other tumours with CIN, and the induction of endoreplication raise the question whether other tumour types with CIN present a similar cell cycle switch.

## 2.4. Centrosome amplification

Centrosomes are dynamic organelles that duplicate once per cell cycle and function as the main microtubule organizing centres during mitosis. Each centrosome comprises two cylindrical structures known as centrioles. In early mitosis, the kinase Polo becomes active and phosphorylates the scaffold protein Centrosomin (CNN). As a consequence, CNN assembles scaffold-like structures and recruits pericentriolar material. Ultimately, γ-tubulin ring complexes are recruited initiating the nucleation of the mitotic spindle. Abnormal centrosomal dynamics can lead to defects in cell division and aneuploidy, and consistently, different centrosomal aberrations are common in human cancer [66].

Centrosome amplification is frequently observed in human tumours and may correlate with poor patient prognosis [67]. However, it is debated whether centrosome amplification is sufficient to cause tumorigenesis. The Sak kinase (SAK) is involved in centriole biogenesis. SAK overexpression leads to supernumerary centrosomes, and this genetic context has been used to study the effects of extra centrosomes in flies. Flies upregulating SAK develop slowly yet they are viable and fertile [68]. Symmetric dividing cells in these animals initially form multipolar spindles but eventually become bipolar, leading to proper chromosome segregation. In this context, mitosis progresses at a slower rate, and this delay is mediated by the SAC, which provides additional time for cells to cluster their centrosomes correctly thus supporting the completion of error-free mitosis. In good agreement, SAC depletion in symmetrically dividing cells with extra centrosomes results in centrosome instability and the formation of multipolar mitosis [68]. By contrast, the induction of supernumerary centrosomes in neuroblasts leads to abnormal mitosis. Asymmetric division in normal neuroblasts produces a new neuroblast and smaller daughter cells that will enter a differentiation path [69]. Intriguingly, some neuroblasts with excessive centrosomes divide symmetrically resulting in brains with an increase in the number of stem cells. These brains develop tumours when injected in healthy fly hosts [68]. Similarly, ectopic centrosomes in ISCs trigger tissue dysplasia [25]. In summary, these studies show that centrosome amplification can drive tumorigenesis in fly stem cells. It is worth noting that, as observed when modelling CIN in flies, cells with ectopic centrosomes trigger different cellular responses in a cell type-dependent manner, and the cellular outcomes of these defects are markedly different between symmetric and asymmetric dividing cells.

Consistent with analyses performed in *Drosophila*, a recent study shows that extra centrosomes can be an initiating event of cancer in mammals [70]. Overexpression of the regulator of centrosome duplication Polo-like kinase 4 (PLK4) leads to supernumerary centrosomes and is a well-established tool to induce centrosome amplification. Mammalian cells overexpressing PLK4 undergo cell cycle arrest in a p53-dependent manner but, in p53-deficient mice, it accelerates tumorigenesis [71–73]. PLK4 was expressed in modest levels in mice over eight months to address whether centrosome amplification can be a single initiating factor. In this condition, chronic centrosome amplification and aneuploidy is observed in several tissues, including the skin, intestine and thymus, as well as the spleen, and mice develop spontaneous tumours, including carcinomas, lymphomas and sarcomas. Tumours caused by chronic PLK4 activation show different karyotypes and high heterogeneity of the tissue, which indicates that tumours undergo continuous errors in chromosome segregation. Notably, the p53 response seems to be generally compromised in these tumours, providing further data supporting that p53 limits the expansion of cells with ectopic centrosomes *in vivo* [70].

Supernumerary centrosomes, independently of its consequential impact in the generation of aneuploidy, can trigger invasion [74]. Centrosome amplification can promote changes in the cellular morphology, and epithelial cells overexpressing PLK4 form dynamic protrusions, lose cell–cell adhesion and invade surrounding tissues in organotypic culture. Mechanistically, centrosome amplification increases microtubule nucleation that in turn increases Rac1 activity, a central regulator of cell–cell adhesion and invasion [74,75]. These results reveal an aneuploidy-independent component of chromosomal amplification in cancer.

## 2.5. Acentrosomal cells

Even though centrosomes are the primary organizers of the mitotic spindle in animal cells, centrosomes are not essential for cell division in *Drosophila*. Flies that lack genes for centrosome maintenance, such as *Spindle assembly abnormal 4* (*Sas-4*) and *asterless* (*asl*), develop into morphologically normal adults [76,77]. Epithelial cells in *Sas-4* mutants use alternative

royalsocietypublishing.org/journal/rsob   Open Biol. **10**: 200060

non-centrosomal spindle assembly mechanisms, including chromatin-mediated microtubule assembly and the SAC, to segregate their chromosomes [78]. Although not essential, centrosomes are required for effective spindle formation and chromosome segregation in epithelial cells. Cells without centrosomes present frequent defects in these processes that lead to JNK-induced apoptosis. Centrosomes are also important to orient symmetric cell divisions in the wing epithelium, and defects in spindle alignment result in delamination and apoptosis [78]. Interestingly, suppression of apoptosis in cells with defects in the alignment of the spindle is sufficient to trigger epithelial–mesenchymal transition (EMT) and tumorigenic growth [79].

Allograft transplantation assays have been used to determine the tumorigenic potential of acentrosomal cells dividing asymmetrically. Neuroblasts mutant for crucial components of centrosome function, including *Sas-4*, *polo* and *aurora A* (*aurA*), develop aneuploid tumours that show constant growth when transplanted into adult flies and can be maintained after multiple allograft rounds [42]. Contrasting to the wing epithelium, centrosome loss in the developing brain does not activate the apoptotic response. In this context, the SAC allows cells to survive and proliferate, and apoptosis is only detected in brains that simultaneously lack centrosomes and central SAC components, as observed in a *Sas-4*, *Mad2* double-mutant background. This genetic context disrupts brain development severely resulting in reduced brain size and lethality [80].

*aurA* encodes a protein kinase involved in neuroblast self-renewal that functions as a tumour suppressor in *Drosophila*. *aurA* mutant neuroblasts display cell fate mis-specification and develop tumours [81,82]. Similarly, *Sas-4* mutant neuroblasts show fate defects, and the resulting tumour phenotype is comparable in both conditions. While the SAC is required to sustain cell viability and proliferation in *Sas-4* mutants, tumour growth is not affected when the SAC is inactive in *aurA* brain tumours, where tumour cells complete mitosis in a SAC-independent manner [81]. Cyclin B is a central cell cycle regulator. The activation of the cyclin-dependent kinase-1 (Cdk1) requires the binding of Cyclin B to induce entry of mitosis, which finishes when Cyclin B is degraded by the activity of the APC/C resulting in Cdk1 inactivation. The activated SAC inhibits the capability of APC/C to ubiquitylate Cyclin B until metaphase [83]. In *aurA* brain tumours, and contrary to *Sas-4* mutants, Cyclin B degradation is delayed leading to prolonged mitosis in a SAC-independent manner, thus avoiding the deleterious effect of SAC malfunction [81]. Therefore, the requirement of the SAC differs in *aurA* and *Sas-4* mutant genetic backgrounds.

As observed in cells with defects in the SAC, centrosome loss leads to the accumulation of ROS [22,84]. ROS is a well-established activator of JNK signalling and may function as an initiating factor of this stress response [85]. Several genes involved in the redox balance are upregulated upon centrosome loss [84]. In this context, glucose-6-phosphate dehydrogenase (G6PD) has been identified as an enzyme that buffers the increase of ROS and that protects cells against cell death. G6PD generates NADPH that can be used to produce glutathione, which functions as a potent antioxidant and hence limits the generation of ROS. The depletion of G6PD in acentrosomal cells results in a robust increase in apoptosis. Therefore, upregulation of G6PD in cells without

centrosomes dampens ROS limiting cellular damage and preventing cell death [84]. In a context of defective SAC, cells trigger the DDR, which buffers CIN-induced aneuploidy and tumorigenesis, and p38, which is required to counteract CIN-induced JNK activation [22] (figure 2*a*).

## 2.6. Cohesin complex errors

The cohesin protein complex has pleiotropic roles. It functions as a molecular glue that holds sister chromatids together after DNA replication. It is also involved in DNA damage repair and plays important roles regulating gene expression. The degradation of these proteins is required for sister chromatid separation in anaphase, which is central for proper chromosome segregation [86]. Given its importance in mitosis, disruption of the cohesin complex can lead to errors in chromosome segregation and aneuploidy. Based on this principle, a genetic tool to induce aneuploidy in a short time window has been developed in *Drosophila* [87]. The system is based on a modified version of the RAD21 cohesin subunit that contains Tobacco Etch Virus (TEV) protease cleavage sites [88]. Upon heat shock induction, the TEV protease cuts RAD21 and causes a dysfunction of the cohesin complex. The system is reversible and, immediately after cohesin cleavage, expresses a TEV-resistant RAD21 protein that restores cohesin activity. Live imaging shows that the functionality of the cohesin complex is completely restored within three rounds of mitosis after heat shock. Therefore, this system allows acute and time-controlled induction of aneuploidy by targeting cohesin [87].

Induction of aneuploidy during larval development by cohesin manipulation is non-lethal, and animals develop into adults. However, they show severe motor defects and reduced lifespan due to the presence of aneuploid cells. The induction of CIN leads to high levels of aneuploidy, where up to 32 chromatids per cell can be observed, and causes neuroblast loss. Although decimated, the neuroblast population is not completely eliminated, and the remaining stem cells continue proliferating through development, which allows a karyotype examination in these cells. *Drosophila* has only two pairs of large chromosomes (chromosomes II and III), one pair of minuscule autosomes (chromosome IV) and a pair of sex chromosomes (X and Y) that are about half the size of the large autosomes. A karyotype analysis shows that loss of the major chromosomes (X, II and III) is not present in these neuroblasts revealing the presence of karyotype restriction, where loss of any of the main chromosomes is incompatible with neuroblast proliferation. The induction of aneuploidy by cohesin depletion in neuroblasts triggers a delayed p53-dependent stress response that can only be detected around 24–48 h after cohesin knockdown. This reveals a delayed response that allows continuous neuroblast proliferation and brain growth, despite the presence of gross aneuploidies [87].

# 3. Stress responses to aneuploidy

The most likely mechanism implicating CIN in tumorigenesis is the generation of aneuploid cells, as a result of chromosome mis-segregation [70]. Although large-scale imbalances are suggested to trigger excessive proliferation in *Saccharomyces cerevisiae* [89], aneuploidy impairs cell fitness and

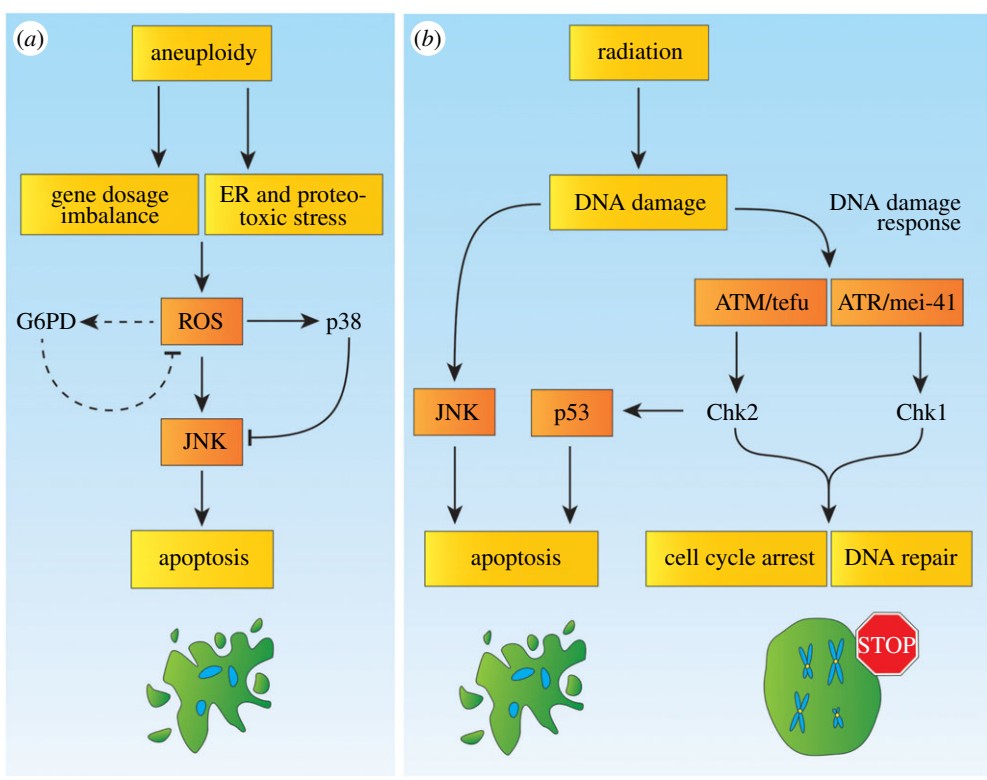

**Figure 2.** Stress responses to aneuploidy and radiation. (*a*) Aneuploidy triggers both gene dosage imbalances and ER as well as proteotoxic stress. This leads together to oxidative stress and ROS accumulation. In turn, ROS can trigger JNK stress signalling. In parallel, cells activate p38 signalling, which can reduce the activity levels of the JNK pathway. In acentrosomal cells, the accumulation of ROS leads to induction of G6PD. This mechanism may also be present in aneuploid cells. It is likely that cells buffer JNK levels through p38 and potentially through G6PD to promote cytoprotective processes first, while higher levels of ROS and JNK activate apoptosis. (*b*) Radiation can trigger DNA damage. This activates primarily the DDR through the ATM/tefu–Chk2 and ATR/mei-41–Chk1 signalling branches. While both contribute to a genotoxic stress response through DNA repair, ATM/tefu–Chk2 mediates mostly p53-mediated apoptosis and can induce G2 cell cycle arrest after low-dose irradiation. ATR/mei-41–Chk1 signalling triggers G2 cell cycle arrest after strong irradiation. Additionally, the JNK stress signalling pathway is activated due to DNA damage. Both p53, which is activated through the DDR, and JNK mediate an apoptotic response in cells with high levels of DNA damage after irradiation. Similar to stress responses to aneuploidy, the involved pathways likely buffer apoptotic signalling to determine whether cells undergo cell cycle arrest and DNA repair or undergo cell death.

proliferation. Accordingly, aneuploidy has severe consequences in animal growth as well as development, and whole organism aneuploidy leads, in most cases, to early lethality [28–31].

Cells with abnormal karyotypes present changes in gene copy number that normally result in alterations in the cell transcriptome and the proteome. This can disturb the normal stoichiometry of protein complexes, hence disrupting cellular functions [31]. Cells use different strategies to balance the negative effect of aneuploidy. Studies in yeast and human cells have revealed that aneuploid cells use protein aggregation mechanisms as an efficient strategy to dampen the excess of subunits involved in protein complexes. This operates as a form of dosage compensation that allows cells to cope with protein complex subunits present in abnormally high proportions [90]. Human aneuploid cells also decrease DNA and RNA metabolism, increase energy and membrane metabolism, and upregulate autophagy as additional means to counteract excessive protein accumulation [91]. Despite the activity of those protecting mechanisms, aneuploidy ultimately impairs cellular functions at different levels. In some cases, it can even lead to a further increase in GIN accelerating the mutational rate and hence enhancing the detrimental impact of aneuploidy [92–94]. The results showing that aneuploidy reduces cellular fitness and impairs cell division seem to conflict with the fact that aneuploidy is typically associated

with cancer, which frequently involves uncontrolled cell proliferation. The boost of GIN, a well-accepted hallmark of cancer, observed in aneuploid cells can partially explain this contradiction. GIN could facilitate the appearance of specific mutations capable of driving cell adaptation, and ultimately malignancy as well as cancer. Aneuploidy enables cellular heterogeneity and fosters cell competition in tissues. This can lead to a microevolution in the tissue and an expansion of clones that can outcompete the surrounding cells [60].

*Drosophila* has been used as an *in vivo* platform to study the consequences of CIN and aneuploidy on cellular fitness and to determine the cellular responses produced to offset these insults. A study based on the manipulation of the sex-specific dosage compensation mechanism (DCM) has recently shown that gene dose imbalance reproduces at least part of the behaviour observed in aneuploid tissues [22]. In *Drosophila*, where a single X chromosome defines males and a set of two X chromosomes defines females, the DCM ensures similar levels of expression in genes located on the X chromosome in both sexes. To compensate this imbalance, the male-specific lethal complex upregulates the expression of genes on the X chromosome. The manipulation of the DCM can be used to induce X chromosome-associated gene dose imbalance. Given that the X chromosome constitutes approximately a fifth of the fly genome, global changes in the expression of this chromosome result in a wide gene dose

royalsocietypublishing.org/journal/rsob    Open Biol. **10**: 200060

imbalance. Experiments using this approach show that male-specific depletion of *Msl2* leads to similar effects to the ones observed when inducing CIN by knocking down elements of the SAC, namely, ROS production, JNK activation as well as apoptosis and tumour formation when cells are apoptosis-resistant [22,23]. The results presented in this study suggest that gene dose imbalance suffices to explain, at least in part, the deleterious effects that CIN has on cell fitness and the pro-tumorigenic potential of aneuploid cells [22] (figure 2*a*).

Other analyses in flies have shown that cells with CIN are exposed to diverse kinds of stresses that sensitize cells to additional insults [95,96]. For example, metabolic disruption at levels well tolerated in normal cells leads to oxidative stress, DNA damage and eventually apoptosis in cells with CIN [96]. These cells also show chaperone upregulation, a characteristic of the endoplasmic reticulum (ER) stress response, and experience difficulties solving additional stress induced by other proteins that have the tendency to aggregate, as proteins with polyQ domains. Moreover, these cells generate ROS and enzymatic disposal of hydrogen peroxide alleviates protein aggregation, indicating that the proteotoxic stress response is intensified by the presence of ROS. In addition to this, CIN also sensitizes cells to nucleotide deprivation inducing additional DNA damage and GIN [95]. These studies expose CIN-associated vulnerabilities that could be exploited to design new therapeutic routes to treat aneuploid tumours (figure 2*a*).

As observed in human cells, *Drosophila* epithelia with CIN activate autophagy [91,97]. Lysosomal activity and autophagy are crucial to cope with aneuploidy and ROS, and inhibition of these processes results in increased apoptosis. In this context, lysosomes are also required to buffer other assaults, such as reduced nucleotide synthesis. As an additional mechanism to counteract the accumulation of ROS, aneuploid cells also use the secretory pathway to eliminate damaged mitochondria [95,97]. A recent report shows that lymphoblastoid cells in human patients with Down syndrome, a disorder caused by trisomy on chromosome 21, show increased hydrogen peroxide levels, chaperones and higher expression of ER stress genes [98]. These results insinuate that the findings obtained in *Drosophila*, rather than being insect-specific, could reflect a universal cell response to aneuploidy.

## 4. Chromosome instability and its implications in invasion and metastasis

Comparative studies have shown that metastatic lesions show higher levels of CIN than primary tumours [99]. This insinuates that CIN contributes to the evolution of malignancy and suggests that CIN may be sufficient to promote invasive and metastatic behaviour. Studies in the wing imaginal disc of *Drosophila* show that cells with CIN protected against apoptosis exhibit invasive traits [100]. These cells emit actin protrusions and transition towards the acquisition of mesenchymal-like morphology. The formation of actin-enriched filopodia is triggered through increased Myosin II and *Drosophila* Filamin—both actin cross-linking proteins. The invasive behaviour relies on JNK signalling, which increases Filamin protein, and the EGFR pathway that stimulates invasiveness through the

induction of ERK and consequent repression of the transcription factor Capicua [100].

The identification of cytosolic nucleic acids is a central aspect in the recognition of pathogens by innate immunity. This leads to the activation of sensor receptors, which in turn mediate host defence mechanisms [101]. Chromosome mis-segregation in cancer cells can lead to micronuclei as well as cytosolic DNA and trigger inflammatory responses, which is suppressed when CIN is inhibited [102]. Cytosolic DNA activates the synthase cGAS and triggers the production of cyclic guanosine monophosphate–adenosine monophosphate (cGAMP), which binds the protein STING. Consequently, the transcription factors IRF3 and NF-κB are activated to mediate an inflammatory response [103,104]. While this cascade triggers interferon I production that mediates a host defence mechanism against viral and bacterial infections, cancer cells are able to hijack this system for their benefit. Ablation of STING in cancer cells leads to reduced EMT gene expression and causes lower inflammatory signalling. Conversely, providing cGAMP to cancer cells with inhibited CIN results in an increased invasive behaviour and metastasis [105]. Together, this study indicates that CIN and the resultant cytosolic DNA trigger immune responses that benefit metastatic behaviour of cancer cells.

Notably, an inflammatory response after cytosolic DNA recognition through STING is conserved in *Drosophila* [106]. STING binds cyclic dinucleotides and triggers an immune response through the immune deficiency pathway that activates the NF-κB protein Relish, which induces the expression of antimicrobial peptides. In agreement with these results, the depletion of STING causes a higher susceptibility to *Listeria* infections. Consistently, overexpression of STING leads to Relish activation and the induction of antimicrobial peptides, which increases the resistance of flies to pathogen infection [106]. This conserved mechanism may be activated after CIN in *Drosophila* as well and could resemble a mammalian response.

## 5. DNA damage response

Cells are exposed to DNA-altering agents in their environment. Exogenous factors (such as UV light and ionizing radiation in the sunlight), genotoxic agents (as for example in cigarette smoke) or medical treatments can cause DNA damage. Additionally, endogenous factors, including DNA replication errors or ROS, can damage the DNA. To combat these insults, cells have evolved DDR mechanisms aimed at maintaining the DNA sequence and structure. However, these processes are not flawless and DNA damage can gradually accumulate over the lifespan of an organism. In fact, malfunction in the DDR is a main source of GIN that accelerates the accumulation of mutations with obvious detrimental effects for the organism [107].

The primary steps of the DDR are to detect the presence and type of lesion in the DNA, halt the cell cycle when needed and, depending on the kind of damage present, activate the necessary repair mechanism. When DNA damage is overly strong, the DDR opts to induce senescence or apoptosis. Initially, the DNA damage is recognized by sensor proteins. The kinases ATM and ATR, *tefu* and *mei-41*, respectively in *Drosophila*, are two main mediators in DNA damage sensing and repair [108]. Studies in mammals show that ATR

royalsocietypublishing.org/journal/rsob    Open Biol. 10: 200060

and ATM induce the DDR through the induction of Chk1 and Chk2, respectively, which are the main mediators of the DDR [109,110].

Cells use different mechanisms to fix lesions in their DNA. Small base damages, including mismatching base pairs and chemical changes of DNA bases, are repaired by the mismatch and base excision repair. Larger lesions in the DNA are fixed by the nucleotide excision repair. The single-strand break repair fixes breaches of a single strand. The most threatening lesions are double-strand brakes (DSBs), which are recognized by phosphorylation of the H2AX histone through ATM, and repaired during G2 through the error-free mechanism called homologous recombination (HR) or during G1 by the error-prone nonhomologous end joining (NHEJ) [109,111].

Oncogene-induced DNA damage is a central source of GIN in cancer [7]. Human tumours frequently carry activating mutations in the oncogene Ras [112], which induces premature entry in S-phase and replicative stress that results in GIN. Despite its prevalence in human cancer, drugs targeting this oncogene in an efficient manner are starting to emerge only recently [113,114]. *Drosophila* has been used to determine the molecular mechanisms by which oncogenic Ras induces GIN and to identify therapeutic and genetic strategies to selectively eliminate Ras-driven tumours [115]. Oncogenic Ras induces replicative stress and DNA damage. At the same time, it activates ERK that inhibits cell cycle arrest and p53-induced apoptosis. Tumour cells activating Ras use ERK activity to survive and accumulate GIN, and consistently, genetic or chemical repression of ERK, combined with ionizing radiation, can efficiently eliminate Ras-induced tumours.

A recent analysis in *Drosophila* involves AurA in the DDR [116]. *aurA* mutant neuroblasts, in addition of showing chromosome aberrations, display inefficient DNA repair and increased sensitivity to ionizing radiation. Genetic analyses in these mutants find that AurA regulates the DNA ligase Lig4 negatively, which is involved in the final steps of the NHEJ repair system. In the light of these results, *aurA* mutants would present enhanced Lig4 activity contributing to the appearance of chromosome aberrations. Additionally, AurA cooperates with Rad51, an enzyme assisting in the resolution of DSBs in post-replicative repair [116]. These results suggest that the negative impact that reduced AurA activity has on genome integrity should be carefully contemplated in therapies involving drugs used to target this kinase.

## 5.1. Radiation and DNA damage response

Ionizing radiation-induced DNA damage is widely used to study the DDR in *Drosophila*. Upon irradiation, different responses have been described in flies. In first place, ATR/mei-41 and ATM/tefu are triggered to activate their downstream targets Chk1 and Chk2, respectively. The ATR/mei-41–Chk1 axis mediates, in large part, G2 arrest and can induce mild p53 induction [117–119]. The ATM/tefu–Chk2 axis is mainly responsible for p53-dependent apoptosis and can induce cell cycle arrest upon low-dose irradiation [120–122]. Additionally, JNK activation triggers apoptosis in a p53-independent manner [123] (figure 2b).

Irradiation of wing imaginal epithelial cells causes tumorigenic growth when apoptosis is suppressed, and this phenocopies the behaviour of cells with an impaired SAC [23]. This tumorigenesis is triggered after irradiation by JNK stress signalling. Strikingly, the DDR functions as a tumour suppressor mechanism in this condition. The study indicates that HR DNA repair in G2 is crucial to suppress tumour growth, while error-prone NHEJ DNA repair during G1 has no effect on tumorigenesis. This gives particular relevance to a G2 cell cycle arrest as part of the DDR. Indeed, a G2 arrest is crucial to suppress tumour growth and the maintenance of this arrest, for example through the inhibition of the G2/M regulator *cdc25/string*, and suppresses tumorigenic growth substantially. Notably, the study indicates both ATR/mei-41 and Chk1 buffer JNK activity in the tissue [124]. As *cdc25/string* is suggested as an important target of JNK stress signalling in tumour suppression [46], the DDR and JNK signalling may determine together whether a cell undergoes a G2 arrest to trigger HR DNA repair or is terminally damaged and undergoes apoptosis (figure 2b).

A short ionizing radiation pulse is sufficient to cause long-lasting JNK activity in fly epithelia [125]. When apoptosis is blocked in irradiated tissues, JNK induces the production of ROS, which in turn reinforces JNK activation and establishes an amplification loop that consolidates this response. A direct consequence is the sustained expression of JNK pro-proliferative targets, such as Wg, JAK-STAT and Dpp, which all together stimulate robust tissue overgrowth [125]. Furthermore, cells express the matrix metalloprotease *Mmp1* that degrades the basement membrane and enables cells to invade other organs [124,126]. Given that HR is crucial to repair lesions of the DNA, tissues that are irradiated and prevented against apoptosis show stronger tumorigenic traits when HR is non-functional [124].

*Drosophila* follicle cells have been used to study the DDR in polyploid cells. During normal development, follicle cells switch to endoreplication in a Notch-dependent manner. After irradiation, the DDR is induced in these polyploid cells. Notably, the downstream apoptotic response is blocked in polyploid cells, and this leads to the accumulation of DNA damage in the tissue [127,128]. These studies reveal a mechanism by which endoreplication and polyploidy may contribute to GIN and tumorigenesis.

# 6. Ageing

Cells replicate and repair their DNA numerous times during the life of an organism. Polymerases copy the DNA with high fidelity, but it is estimated that they incorporate erroneous nucleotides once per $10^8$–$10^{10}$ nucleotides polymerized [129]. Furthermore, the DDR is not totally efficient. For example, NHEJ is intrinsically inaccurate. Moreover, DNA repair mechanisms are impaired with age, which causes an age-related increase in the mutation rate [130–132]. Together, these defects cause the continuous accumulation of alterations in the DNA, which is considered a central molecular driver of ageing [133]. An additional consequence is the presence of high levels of cell heterogeneity in aged tissues. This can lead to competitive interactions between different cells, hence facilitating the propagation of pre-malignant lesions in early stages of tumour formation [60,107].

## 6.1. DNA damage response and lifespan

The observation that the DDR becomes compromised with age, leading to a higher frequency of mutations, opens the

royalsocietypublishing.org/journal/rsob    Open Biol. **10**: 200060

**Figure 3.** Loss of chromosomal integrity and accumulation of mutations. (*a*) Ageing fly intestinal epithelia show frequent and spontaneous dysplastic growth arising from the intestinal stem cell population (green). These stem cells show frequent large-scale rearrangements in their chromosomes with increased age. In particular, the *Notch* locus, which is localized on the X chromosome, is frequently impaired and undergoes a loss heterozygosity leading to tumorigenic growth. Male flies in particular show frequent dysplasia due to the presence of one X chromosome. (*b*) The oncogene-induced DNA damage model describes how an initial oncogene activation can trigger broad DNA damage implications. In the first place, oncogenes, such as Ras or Cyclin E, can trigger DNA replication stress when the chromosomes are replicated during the cell cycle in S-phase. This can trigger DNA damages, and especially common fragile sites are susceptible to breaks and rearrangements. Additionally, tumorigenesis triggers copy number variations and single-nucleotide polymorphisms. Together, these processes can lead to an erosion of chromosomal integrity and broad changes to the encoded genetic information.

question whether upregulation of DDR genes may increase lifespan. Genes involved in the DDR were overexpressed in *Drosophila* to test this hypothesis, and the effect on lifespan as well as stress resistance to different insults, such as hyperthermia, oxidative stress and starvation, was analysed. The genes tested included parts of the main DDR machinery, base excision repair, nucleotide repair and DSB repair. The obtained results regarding lifespan and stress resistance varied between the different conditions analysed. Importantly, the expression of *hus1* and *chk2*, which are involved in the main DDR; *mei-9* and *mus210*, which are part of the nucleotide excision repair; and *WRNexo*, which plays a role in DSB repair, are able to extend the lifespan and increase stress resistance in the animals analysed. This provides evidence supporting that upregulation of specific DDR elements can have positive effects in ageing [134].

DNA repair defects are associated with syndromes that present premature ageing, such as Werner syndrome. This condition results from mutations in WRN, a gene encoding for an essential enzyme involved in DNA repair [135]. The analysis of WRN-deficient flies revealed that these animals are sensitive to replicative stress, show shorter lifespan and present increased tumour incidence [136]. This work provides additional evidence substantiating the link between defective DNA repair with premature ageing and cancer. Besides, it provides a tractable *in vivo* model to study the molecular basis of DNA damage-related ageing.

## 6.2. Spontaneous tumorigenesis in ageing flies

Spontaneous tissue dysplasia has been observed in the midgut of aged flies. The fly midgut is a well-organized epithelium maintained by the activity of ISCs. During the ageing process, the epithelium accumulates ISCs and shows morphological irregularities. In the midgut, the JNK pathway coordinates tissue regeneration by inducing proliferation in ISCs. However, the effects of JNK activation in response to stress vary in an age-dependent manner. While stress-induced JNK drives ISC proliferation and regeneration in young flies, JNK signalling in aged individuals results in the accumulation of ISCs and defects in the epithelial architecture of the midgut [137]. Compared with young flies, ISCs in aged individuals show regular loss of heterozygosity caused by HR and frequent rearrangements that cause neoplastic growth. The majority of age-related neoplastic lesions correlate with the presence of deletions and complex rearrangements in the locus of the gene *Notch* [138] (figure 3*a*).

The presence of invasive cancers in humans increases from age 40 on [139]. Although this time exceeds significantly the average lifespan of *Drosophila*, a comparative analysis between ageing flies and mice revealed a higher frequency of mutations in flies than in mice, suggesting that the mutation rate is proportional to the biological, rather than the chronological age [140,141]. This might provide an

explanation as why *Drosophila* can develop tumours during its relatively short lifespan.

## 6.3. Drugs, tumours and healthy ageing

ISCs have been used as a model to study the effects of the drug metformin in healthy ageing. Metformin is among the most prescribed medications in developed countries. This drug is used to treat type 2 diabetes and acts by reducing insulin resistance and plasma insulin concentration, causing a reduction in blood glucose levels. Metformin is also emerging as a method for the prevention and treatment of cancer [142]. ISCs in old flies are exposed to oxidative stress and DNA damage, and aged flies can accumulate ISCs and develop tumours [137,138,143,144]. Treatment with metformin reduces ROS and DNA damage in ISCs, and this inhibits the formation of hyperplasia in aged flies [145]. Consistently with these results, metformin prolongs lifespan in mice [146]. These studies provide additional evidence, suggesting that reducing DNA damage can improve healthy ageing.

Flies have been also used to interrogate the interactions between metformin and other anticancer drugs. Doxorubicin is used to treat cancer and acts by targeting and reducing the activity of topoisomerase 2, which is required for tumour cell proliferation. Despite its anti-tumoural effect, its use has adverse consequences and induces cardiotoxicity presumably due to the induction of ROS and DNA damage [147]. Experiments in flies show that metformin alleviates the negative effects of doxorubicin [148]. While doxorubicin on its own induces DNA breaks and the formation of epithelial tumours in *Drosophila*, the combined administration of doxorubicin with metformin results in reduced DNA damage and tumour incidence, demonstrating that metformin alleviates the negative effect of doxorubicin in this experimental condition [148]. The specific molecular function of metformin in this context has not been determined and needs to be investigated.

## 7. Accumulation of mutations

GIN promotes the accumulation of different types of mutations. However, the presence and pattern of mutations in different tumour types do not occur entirely at random. Some regions of the genome are more fragile and susceptible to accumulate errors. Besides, specific factors can determine the pattern of mutations present in different cancers. For example, some tumours accumulate specific genomic changes that vary depending on the initial oncogenic factors [149].

## 7.1. Chromosomal fragile sites

Chromosomal fragile sites are unstable loci that preferentially show gaps and breaks in metaphasic chromosomes when cells are exposed to replication stress. Over 120 individual chromosomal fragile sites have been described in humans [150]. They are subdivided into two categories: common fragile sites (CFSs) and rare fragile sites (RFSs). RFSs occur in few individuals—below 5% of the population—and follow Mendelian inheritance. In most cases, an expansion of repetitive nucleotide sequences is a central factor mediating the manifestation of RFSs. CFSs represent the majority of fragile sites. They are well conserved trough evolution and are molecular hotspots in cancer. CFSs are typically late-replicating large genes with several exons that have a reduced number of origins of replication. Moreover, they contain relatively high AT-rich sequences that lead to the formation of secondary structures. These features can together predispose the DNA to the appearance of physical collisions between the transcription and replication machineries affecting the stability of these regions and eventually causing chromosome breaks [151,152].

Fragile sites induce GIN and are associated with human diseases, such as cancer. *In vitro* studies show that breaks at CFSs occur after exposure to low doses of aphidicolin, an inhibitor of DNA replication that induces replicative stress [153]. DNA polymerase δ is a highly conserved enzymatic complex that plays central roles in DNA replication and repair in eukaryotes [154]. In its basic form, this complex includes PolD, the catalytic subunit; Pol31, a structural subunit; and Pol32, an auxiliary subunit [155]. A recent study in *Drosophila* involves Pol32 in the generation of CFSs and shows that, although Pol32 is crucial for genome replication in early development, loss of Pol32 in later developmental stages does not impede DNA replication and, instead, induces the expression of CFSs [156] (figure 3*b*).

Oncogene activation can induce replicative stress and specific fragile site landscapes that correlate to common breakpoints in cancer [149]. Thus, oncogene activation can facilitate GIN at fragile sites, which can potentially influence other hallmarks of cancer. Chromosomal breaks at CFSs can have oncogenic consequences. For instance, FRA3B and FRA16D are CFSs that are located within tumour suppressor gene loci. The tumour suppressor gene FITH is located at the most active CFS, FRA3B, and is commonly dysregulated in cancer. Similarly, the second most active CFS, FRA16D, is situated within the WWOX gene [157]. In some cases, CFSs can promote the amplification of oncogenes, such as MET [158]. In summary, CFSs correlate with chromosome breakpoints commonly found in cancer genomes [159,160] (figure 3*b*).

WWOX is a long gene that resides in the FRAD16D region, which is the second most common CFS in humans. Loss of heterozygosity and homozygous deletions of WWOX are commonly found in cancer [161]. WWOX is an oxidoreductase involved in different cellular processes, including cell metabolism, osteoblast differentiation and steroidogenesis. Additionally, WWOX acts as a tumour suppressor and affects DDR, p53 regulation and apoptosis. WWOX is conserved from insects to mammals [162,163].

*Drosophila* Wwox shows a 49% similarity to the human WWOX protein. Although WWOX null mutant mice show severe metabolic defects and die in early development, null fly mutants show no obvious phenotype [164,165]. *Wwox* mutant viability enabled an *in vivo* screen for Wwox-interacting proteins. This screen identified 27 candidate genes involved in aerobic metabolism. Genetic interactions revealed functional connections with two proteins closely connected with the tricarboxylic acid (TCA) cycle: isocitrate dehydrogenase (Idh) and superoxide dismutase 1 (Sod1). In *Drosophila*, knockdown of *Idh* leads to decreased viability, which is enhanced by *Wwox* depletion. Reciprocally, *Wwox* expression increases the viability of animals with reduced Idh. The TCA cycle produces ROS. Sod1 is an enzyme converting superoxide to hydrogen peroxide playing a crucial role in

the ROS detoxification machinery. Although *Sod1* and *Wwox* mutants do not show viability defects on their own, *trans*-heterozygous mutants present reduced viability. Additional experiments suggest that Sod1 regulates *Wwox* transcript levels, both in *Drosophila* and HEK cells. Consistent with these observations, Wwox has been shown to affect ROS levels in *Drosophila* larvae [166].

Studies in mammalian models show that WWOX partners with p53 to control cell death [167]. TNF-α is a central regulator of apoptosis [168], and functional studies show that Wwox cooperates with the TNF-α fly orthologue, Eiger (Egr), in the regulation of the apoptotic response [169]. Egr overexpression in the fly eye disrupts ommatidial morphology and reduces overall eye size leading to a characteristic 'rough eye' phenotype. Remarkably, *Wwox* depletion rescues the Egr-induced 'rough eye' phenotype, and *Wwox* overexpression enhances this defect. Mechanistically, Egr activates Wwox, leading to ROS release and ROS-dependent apoptosis. Finally, Wwox has been shown to protect against ionizing radiation, suggesting an additional mechanisms by which this gene could influence tumour development [165]. In summary, studies in flies have served to identify novel tumour suppressor roles of Wwox [165,166,169].

## 7.2. Copy number variations and single-nucleotide polymorphisms

Cancer develops through the accumulation of mutations. Ultimately, this can lead to aggressive, proliferative and metastatic cells. GIN is thought to accelerate the acquisition of mutations in tumour cells. Indeed, human cancer cells accumulate genomic alterations faster than healthy cells [170,171]. Although *Drosophila* larval neoplasms develop for a short time period of around one or two weeks, tumour allografts into adult flies facilitate long-term analysis of fly tumours. This approach has been used to address whether, as observed in human cancer, *Drosophila* tumours accumulate copy number variations (CNVs), understood as changes in the number of copies of a given gene between different individuals; and single-nucleotide polymorphisms (SNPs), which represent a difference in a single DNA nucleotide at a specific position in the genome [172]. Several well-characterized brain tumours types, such as *l(3)mbt*, *brat*, *aurA* or *lgl*, were sequenced after the first, fifth and 10th round of transplantation. Generally, the different tumours analysed remain close to a diploid karyotype, indicating that aneuploidy is not preferentially selected during tumour evolution in these tumour types. Remarkably, these tumours show CNVs in their genome. The number of CNVs varies between tumour types, whereas *l(3)mbt* tumours show the lowest number with 11 CNVs, *brat* tumours present 80 CNVs after the 10th round of transplantation. In most of the cases, these alterations affect coding regions, and the largest CNVs (above 500 kb) are observed in late rounds of allografting [172] (figure 3*b*).

SNPs are present in all tested tumours. However, as observed in human tumours, the frequency of SNPs varies substantially between the *Drosophila* tumour types. While *brat* tumours seem less susceptible to SNP accumulation, *aurA* and *lgl* tumours present a dramatic eightfold increase in the number of SNPs in late rounds of transplantation. Interestingly, the number of SNPs per Mb is detected in a comparable range in the tested *Drosophila* tumours and human cancer types. Ultimately, this study indicates that, similar to human tumours, *Drosophila* tumours accumulate CNVs and SNPs over time. Furthermore, the different tumours analysed show diverse susceptibility to CNVs and SNPs, which indicates that GIN varies in a tumour type-dependent manner [172] (figure 3*b*).

## 8. Conclusion

Aneuploidy has generally deleterious effects on growth and development. However, most solid cancers show aneuploidies and tumour progression correlates with GIN [173,174]. GIN increases genetic variability and cellular heterogeneity contributing to natural selection during tumour evolution. This selection process can favour the accumulation of specific cancer-enabling mutations and facilitate drug resistance in specific cell subclones [175]. In consonance with this, heterogeneous cell populations have been observed in cancer tissues. A pan-cancer analysis addressing intratumour heterogeneity by determining clonal populations reveals that tumours show on average four clonal subpopulations; 86% of tumours contain at least two subpopulations; and the presence of two or more clones worsened the patient outcome [176]. Although aneuploidy generally impairs cellular fitness, it can be used to increase genetic variability, which can accelerate tumour progression by increasing cell adaptability and by selecting highly competitive cancer cells [177].

The larval imaginal discs are commonly used to model different aspects of tumour formation [27]. Despite its popularity, this model has some obvious limitations in the study of tumour evolution. Genetically induced tumours in *Drosophila* larvae develop for approximately one to two weeks before these animals die. This is a short time window for tumour microevolution and expansion of different clonal subpopulations, as observed in human cancers. Consistent with this, analysis of structural variants in fly larval tumours driven by depletion of the *Drosophila* tumour suppressor Polyhomeotic does not show a loss of genome integrity [178,179]. Nevertheless, this system has proven valuable in investigating the immediate- and short-term effects of GIN in connection with tumorigenesis [22–24,46,124] (figure 4*a*). Tumour transplantation in adult hosts emerges as an ideal alternative to circumvent these limitations allowing to explore the potential changes during tumour evolution present in different genetic contexts [41,172]. This approach has been used to examine CIN in neuroblasts, where cells show a dramatic increase in polyploidy and aneuploidy after several transplantation steps [42,68,81,82] (figure 4*b*). This approach also reveals that fly neoplasms show GIN and accumulate SNPs as well as CNVs that are in the range of human cancers [172]. Additionally, ISCs have been helpful in the study of stem cells and an analysis of ageing in adult flies [25,138] (figure 4*c*).

The primary consequences of aneuploidy are changes in gene copy number and gene dose imbalance. Changes in copy number alter the expression of multiple genes that can result in gain or loss of cancer genes, which would certainly influence tumour progression. Additionally, chromosome imbalance can trigger different cellular responses in the form of activation or inhibition of certain pathways that may contribute to tumour growth and malignancy. Flies

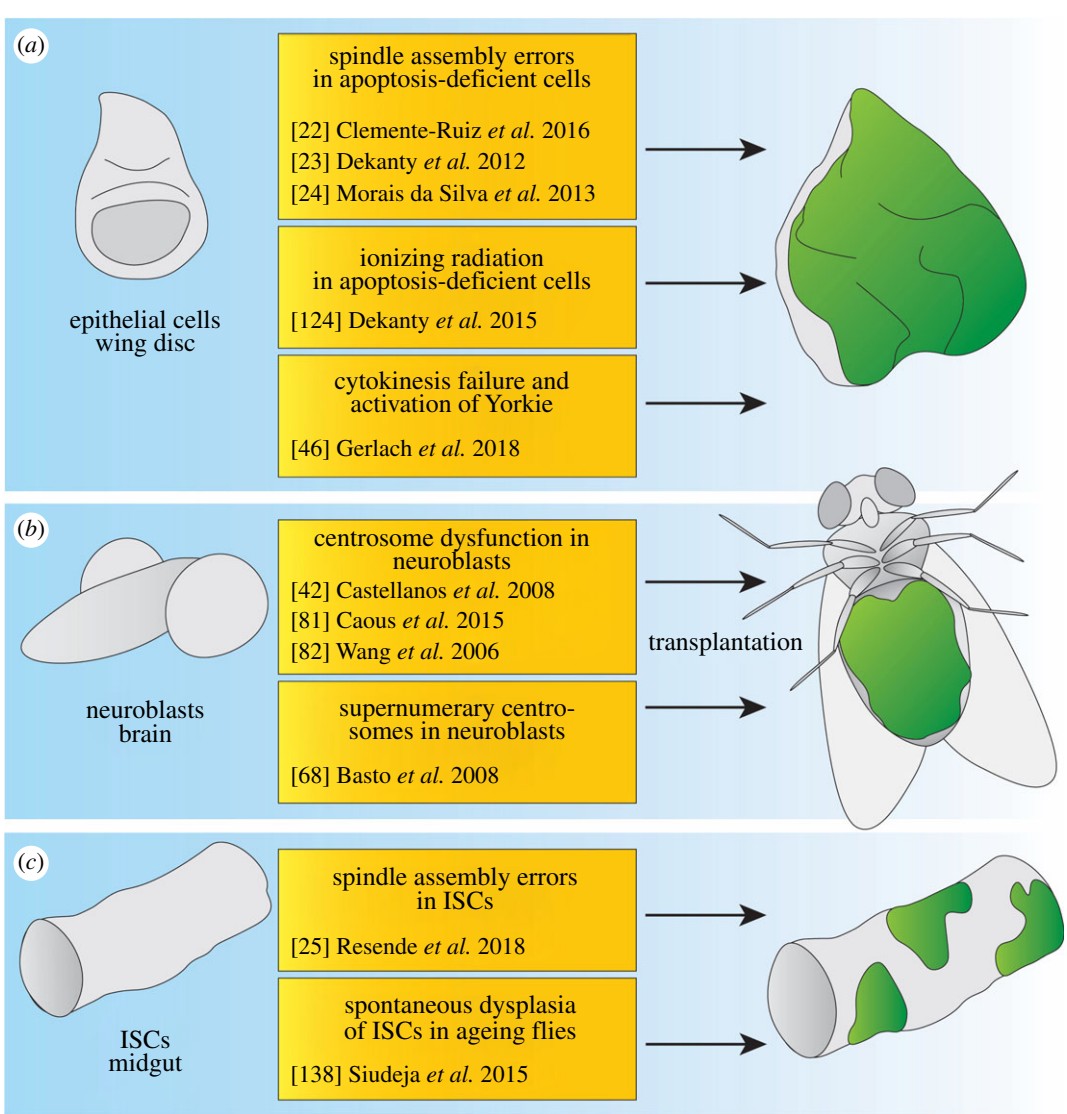

**Figure 4.** Models to study the connection between genomic instability and tumorigenesis in flies. (a) The wing imaginal disc has been used extensively to model aspects of genomic instability in epithelial cells. In this tissue, spindle assembly errors trigger tumorigenesis in apoptosis-deficient cells through the JNK signalling axis. Similar to spindle assembly errors, ionizing radiation triggers neoplastic growth of epithelial cells that are apoptosis-deficient, and this tumorigenesis is enhanced by the depletion of DDR genes. Cytokinesis failure triggers the JNK pathway as well, and an introduction of the oncogene Yorkie leads to tumorigenesis that gives rise to highly polyploid cells. (b) Neuroblasts divide asymmetrically and the presence of centrosomes is essential for error-free mitosis in these cells. The ablation or mis-localization of centrosomes as well as supernumerary centrosomes can lead to tumorigenic neuroblasts. Transplantations of neuroblasts with centrosome defects form tumours in the abdomen of adult flies. (c) ISCs trigger dysplastic growth after spindle errors occur and behave similar to apoptosis-deficient epithelial cells. Similarly, ageing flies trigger frequently dysplasia originating from ISCs, and this is due to age-related somatic mutations.

have their genome packed in only three major pairs of homologous chromosomes. This contrasts with humans, where the genome is distributed in 23 pairs of chromosomes. Segregation mistakes in any of the main chromosomes of the fly will affect a vast portion of the genome, limiting the use of this model system to study the consequences of gaining and/or losing specific genes present in particular chromosomes. Although this could be viewed as a limitation, it allows to study the general consequences of aneuploidy, independently of the specific genes affected by segregation errors. *Drosophila* therefore provides a potent system to determine the cellular responses and consequences of gene dose imbalance in aneuploid cells.

Studies in flies agree that the JNK pathway constitutes a central response to CIN and mediates key cellular responses, such as apoptosis, proliferation and transformation. Specifically, depletion of SAC components, the presence of acentrosomal cells and cytokinesis defects induce JNK-dependent apoptosis in *Drosophila* epithelial cells [23,46,78]. JNK also controls the cell cycle, as observed in cells with defective cytokinesis where JNK inhibits Cdc25/string, and therefore limits G2/M progression [46]. Cdc25/string regulation by JNK has also been reported to operate during wound healing [180]. This regulatory interaction may be conserved and has also been reported in human cells where JNK regulates entry in mitosis by regulating CDC25C [181]. Moreover, JNK inhibits Cyclin B, another central G2/M regulator, in *Drosophila* endoreplicating cells [65]. Furthermore, JNK controls the acquisition of mesenchymal-like characteristics and tumour invasion [100]. Even though JNK plays a central role in response to GIN, the functions played by this signalling pathway appear to be very complex and, while in some contexts it limits tumour progression, in others it behaves as a pro-tumorigenic factor [85,182,183].

The tumour suppressor gene p53 plays a central role in preserving genome integrity. In the presence of different

forms of cellular stress, such as oncogene activation or DNA damage, p53 controls diverse cellular responses to maintain tissue homeostasis. The function of p53 in response to GIN is notably different between flies and mammals. While in mammals p53 induces cell cycle arrest and apoptosis in response to stress stimuli, in flies its function is predominantly limited to the induction of the apoptotic response [184–186]. Additionally, p53 limits the expansion of tetraploid and aneuploid cells after cytokinesis failure and induces apoptosis in these conditions [34,45,187,188]. While p53 is activated by some forms of GIN in flies, the generation of aneuploid cells by manipulation of the SAC or by cytokinesis failure leads to p53-independent and JNK-dependent cell elimination through apoptosis [23,46].

The severity of GIN influences cell behaviour differently and, while moderate GIN can promote mutability and tumorigenesis, strong GIN can be incompatible with cell viability. Supporting this notion, results in mice show that, although weak SAC inhibition is oncogenic, severe SAC disruption increases cell death and suppresses tumour formation [189]. Similar outcomes are observed in cultured glioblastoma cells. These cells have segregation defects, show aneuploid karyotypes and form tumours when injected in mice. Additional induction of CIN through the expression of a microtubule-depolymerizing kinesin, which leads to lower error correction during chromosome segregation, causes tumour suppression, and tumours were not formed when these cells were injected in mice [190]. Consistently, a pan-cancer analysis assessing CNVs shows that, while CNVs affecting less than 25% or more than 75% of the genome is beneficial for patient survival, intermediate CNV levels result in a worse patient outcome [176]. These studies point towards a novel therapy based on the induction of additional GIN to induce lethality and suppress disease progression [191]. These observations are relevant for fly tumour models where different strategies to manipulate gene activity, such as hypomorphic mutants, null alleles or RNAi-mediated gene knockdown, may result in different cellular outcomes.

Data accessibility. This article has no additional data.

Authors' contributions. S.U.G. and H.H. conceived of the review, and designed, coordinated and drafted the manuscript. S.U.G. gave final approval for publication and agree to be held accountable for the work presented therein.

Competing interests. We declare we have no competing interests.

Funding. This work was supported by the Novo Nordisk Foundation (grant no. NNF0052223) and a grant by the Neye Foundation for genetic models for cancer gene discovery.

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
