## [Reviewer comments · Open Biology]

Review History

RSOB-20-0060.R0 (Original submission)

Review form: Reviewer 1

Recommendation

Accept as is

Do you have any ethical concerns with this paper?

No

Comments to the Author

Gerlach and Herranz's review manuscript is an encyclopedic effort to summarize discoveries of the last two decades carried out by the *Drosophila* community (including Herranz's lab) on the contribution of genomic instability (GIN) to cancer. I have nothing to say but congratulate the authors for such a wonderful work which will be of benefit to the whole scientific community interested in GIN and cancer and will contribute to spread the important discoveries carried out but this tiny insect. The ms is well organized, clearly written and thoroughly discussed and figures are self-explanatory. I support this ms for immediate publication in Open Biology. I will be a central review in the field, I am convinced.

Just a few comments on some definitions.

Pag5

"CIN....refers to large-scale genomic alterations that can lead to substantial rearrangements of

chromosomal regions"

This is not completely correct. CIN is the continuous variation in chromosome number and/or structure due to defects in mitosis.

"CIN...can lead to substantial rearrangements of chromosomal regions; polyploidy, a situation where cells contain more than two sets of chromosomes"

CIN does not induce polyploidy. This is usually a consequence of cytokinesis failure.

Pag 7

"Errors that lead to defects in mitosis include ... cytokinesis failure"

I would put these in two different categories and they will give rise to different consequences: mitotic defects to aneuploidy and cytokinesis failure to polyploidy

Pg 7:

"Aneuploidy, defined as the presence of an abnormal number of chromosomes in a cell, has generally a negative effect on cell fitness"

This definition also includes polyploidy. I would say a number of chromosomes which is not a multiple of the haploid karyotype, or unbalanced number of chromosomes...

Pg 8: "Induction of GIN by SAC downregulation"

I would say CIN instead of GIN

Pg 8: "This allograft assay, developed by the group of Cayetano González, has been extensively used to study the oncogenic potential of different genetic conditions in flies." This is not correct. Elizabeth Gateff was the one who first used this technique to address malignancy in fly tumors and this technique was previously and widely used by the Hadorn lab to analyze determination of adult structures. I would include though the merit of Gonzalez's lab to re-discover this technique for the fly community.

Decision letter (RSOB-20-0060.R0)

16-Apr-2020

Dear Dr Herranz,

We are pleased to inform you that your manuscript RSOB-20-0060 entitled "Genomic Instability and Cancer - Lessons from Drosophila" has been accepted by the Editor for publication in Open Biology. The reviewer(s) have recommended publication, but also suggest some minor revisions to your manuscript. Therefore, we invite you to respond to the reviewer(s)' comments and revise your manuscript.

Please submit the revised version of your manuscript within 7 days. If you do not think you will be able to meet this date please let us know immediately and we can extend this deadline for you.

When submitting your revised manuscript, you will be able to respond to the comments made by the referee(s) and upload a file "Response to Referees" in "Section 6 - File Upload". You can use

this to document any changes you make to the original manuscript. In order to expedite the processing of the revised manuscript, please be as specific as possible in your response to the referee(s).

- 1) A text file of the manuscript (doc, txt, rtf or tex), including the references, tables (including captions) and figure captions. Please remove any tracked changes from the text before submission. PDF files are not an accepted format for the "Main Document".
- 2) A separate electronic file of each figure (tiff, EPS or print-quality PDF preferred). The format should be produced directly from original creation package, or original software format. Please note that PowerPoint files are not accepted.
- 3) Electronic supplementary material: this should be contained in a separate file from the main text and meet our ESM criteria (see <http://royalsocietypublishing.org/instructions-authors#question5>). All supplementary materials accompanying an accepted article will be treated as in their final form. They will be published alongside the paper on the journal website and posted on the online figshare repository. Files on figshare will be made available approximately one week before the accompanying article so that the supplementary material can be attributed a unique DOI.

Online supplementary material will also carry the title and description provided during submission, so please ensure these are accurate and informative. Note that the Royal Society will not edit or typeset supplementary material and it will be hosted as provided. Please ensure that the supplementary material includes the paper details (authors, title, journal name, article DOI). Your article DOI will be 10.1098/rsob.2016[*last 4 digits of e.g. 10.1098/rsob.20160049*].

- 4) A media summary: a short non-technical summary (up to 100 words) of the key findings/importance of your manuscript. Please try to write in simple English, avoid jargon, explain the importance of the topic, outline the main implications and describe why this topic is newsworthy.

Images

Data-Sharing

It is a condition of publication that data supporting your paper are made available. Data should be made available either in the electronic supplementary material or through an appropriate repository. Details of how to access data should be included in your paper. Please see <http://royalsocietypublishing.org/site/authors/policy.xhtml#question6> for more details.

Data accessibility section

Sincerely,
The Open Biology Team
mailto:openbiology@royalsociety.org

Reviewer's Comments to Author:
Referee:

Comments to the Author(s)

Gerlach and Herranz's review manuscript is an encyclopedic effort to summarize discoveries of the last two decades carried out by the Drosophila community (including Herranz's lab) on the contribution of genomic instability (GIN) to cancer. I have nothing to say but congratulate the authors for such a wonderful work which will be of benefit to the whole scientific community interested in GIN and cancer and will contribute to spread the important discoveries carried out but this tiny insect. The ms is well organized, clearly written and thoroughly discussed and figures are self-explanatory. I support this ms for immediate publication in Open Biology. I will be a central review in the field, I am convinced.

Just a few comments on some definitions.

Pag5

"CIN...refers to large-scale genomic alterations that can lead to substantial rearrangements of chromosomal regions"

This is not completely correct. CIN is the continuous variation in chromosome number and/or structure due to defects in mitosis.

"CIN...can lead to substantial rearrangements of chromosomal regions; polyploidy, a situation where cells contain more than two sets of chromosomes"

CIN does not induce polyploidy. This is usually a consequence of cytokinesis failure.

Pg 7

"Errors that lead to defects in mitosis include ... cytokinesis failure"

I would put these in two different categories and they will give rise to different consequences: mitotic defects to aneuploidy and cytokinesis failure to polyploidy

Pg 7:

"Aneuploidy, defined as the presence of an abnormal number of chromosomes in a cell, has generally a negative effect on cell fitness"

This definition also includes polyploidy. I would say a number of chromosomes which is not a multiple of the haploid karyotype, or unbalanced number of chromosomes...

Pg 8: "Induction of GIN by SAC downregulation"

I would say CIN instead of GIN

Pg 8: "This allograft assay, developed by the group of Cayetano González, has been extensively used to study the oncogenic potential of different genetic conditions in flies."

This is not correct. Elizabeth Gateff was the one who first used this technique to address malignancy in fly tumors and this technique was previously and widely used by the Hadorn lab to analyze determination of adult structures. I would include though the merit of Gonzalez's lab to re-discover this technique for the fly community.

Author's Response to Decision Letter for (RSOB-20-0060.R0)

See Appendix A.

Decision letter (RSOB-20-0060.R1)

01-May-2020

Dear Dr Herranz

We are pleased to inform you that your manuscript entitled "Genomic Instability and Cancer – Lessons from *Drosophila*" has been accepted by the Editor for publication in Open Biology.

Sincerely,
The Open Biology Team
mailto: openbiology@royalsociety.org

Appendix A

We appreciate the comments by the reviewer. The manuscript has been reviewed accordingly. See the modifications, point by point, below.

-

Reviewer's Comments to Author:

Referee:

Comments to the Author(s)

Gerlach and Herranz's review manuscript is an encyclopedic effort to summarize discoveries of the last two decades carried out by the *Drosophila* community (including Herranz's lab) on the contribution of genomic instability (GIN) to cancer. I have nothing to say but congratulate the authors for such a wonderful work which will be of benefit to the whole scientific community interested in GIN and cancer and will contribute to spread the important discoveries carried out but this tiny insect. The ms is well organized, clearly written and thoroughly discussed and figures are self-explanatory. I support this ms for immediate publication in *Open Biology*. I will be a central review in the field, I am convinced. Just a few comments on some definitions.

Pag5

"CIN....refers to large-scale genomic alterations that can lead to substantial rearrangements of chromosomal regions"

This is not completely correct. CIN is the continuous variation in chromosome number and/or structure due to defects in mitosis.

"CIN...can lead to substantial rearrangements of chromosomal regions; polyploidy, a situation where cells contain more than two sets of chromosomes"

CIN does not induce polyploidy. This is usually a consequence of cytokinesis failure.

This has been corrected. In the reviewed version of the manuscript it appears as:

"CIN is the most frequent type of GIN in human cancers and refers to the changes in chromosome number and/or structure as a consequence of defects in mitosis. The most primary consequence of CIN is the formation of cells with unbalanced chromosome content, a condition known as aneuploidy [12]."

Pg 7

"Errors that lead to defects in mitosis include ... cytokinesis failure"

I would put these in two different categories and they will give rise to different consequences: mitotic defects to aneuploidy and cytokinesis failure to polyploidy

This has been corrected. In the reviewed version of the manuscript it appears as:

"Errors affecting mitosis include malfunctioning of the SAC, inefficient cohesion between sister chromatids, defective attachment between the microtubules and chromosomes, centrosome amplification, incorrect timing of centrosome separation [19]. These defects typically result in aneuploidy. Flaws in cytokinesis, the last step of mitosis, can also occur and cause the formation of polyploid cells [20]."

Pg 7:

“Aneuploidy, defined as the presence of an abnormal number of chromosomes in a cell, has generally a negative effect on cell fitness”

This definition also includes polyploidy. I would say a number of chromosomes which is not a multiple of the haploid karyotype, or unbalanced number of chromosomes...

This has been corrected. In the reviewed version of the manuscript it appears as:

“Aneuploidy, a cell condition defined by an unbalanced number of chromosomes, has generally a negative effect on cell fitness [28-31].”

Pg 8: “Induction of GIN by SAC downregulation”

I would say CIN instead of GIN

This has been corrected. In the reviewed version of the manuscript it appears as:

“Induction of CIN by SAC downregulation, kinetochore malfunction, or centrosomes amplification leads to the accumulation of aneuploid ISCs in the fly intestine.”

Pg 8: “This allograft assay,

developed by the group of Cayetano González, has been extensively used to study the oncogenic potential of different genetic conditions in flies.”

This is not correct. Elizabeth Gateff was the one who first used this technique to address malignancy in fly tumors and this technique was previously and widely used by the Hadorn lab to analyze determination of adult structures. I would include though the merit of Gonzalez's lab to re-discover this technique for the fly community.

This has been corrected. In the reviewed version of the manuscript it appears as:

“This allograft assay has been extensively used to study the oncogenic potential of different genetic conditions in flies. The analysis of mutants for the SAC components bub3 and bubR1 in this assay did not identify the formation of brain tumors, which indicates that the cellular responses to the presence of aneuploidies in stem cells varies depending on the cellular context [42].”